# The Usefulness of Accounting Information and Management Accounting Practices under Environmental Uncertainty

**Rui Pires** [1,*]🆔, **Maria-Ceu Gaspar Alves** [2]🆔 and **Catarina Fernandes** [1,3]🆔

1    Instituto Politécnico de Bragança, 5300-253 Bragança, Portugal
2    NECE-UBI Research Unit in Business Sciences, University of Beira Interior, 6201-001 Covilhã, Portugal
3    Centro de Economia e Finanças da Universidade do Porto (CEF.UP), Faculdade de Economia da Universidade do Porto, 4200-464 Porto, Portugal
*    Correspondence: rucapires@ipb.pt

**Abstract:** The purpose of this paper is twofold. Firstly, we aim to investigate the relationships among environmental uncertainty, broad-scope and timely management accounting information usefulness, and (traditional and contemporary) management accounting practices (MAPs) usage. Secondly, we intend to explore how these relationships influence decision-makers' satisfaction with management accounting information. Survey data were obtained through an online questionnaire from 114 large manufacturing companies operating in Portugal. The findings indicate a positive relationship between environmental uncertainty and timely management accounting information usefulness and between (broad-scope and timely) management accounting information usefulness and (traditional and contemporary) MAPs usage. The findings also show that decision-makers' satisfaction with management accounting information improves when there is a good fit between environmental uncertainty, broad-scope and timely management accounting information usefulness, and MAPs usage. In this way, organisations need to adjust the implementation and usage of MAPs to contextual factors, using both contemporary and traditional MAPs, to achieve greater decision-makers' satisfaction with management accounting information. Thus, the results achieved in this study are useful for both theory and practice and have several implications for professionals engaged in MAPs implementation and decision-making activities.

**Keywords:** environmental uncertainty; management accounting information usefulness; management accounting practices (MAPs); satisfaction with management accounting information; congruent fit; online questionnaire survey; Portugal

## 1. Introduction

Constant changes in the organisational environment, resulting from rapid technological advances, increased competition, high customer expectations, and new demands in terms of social and environmental responsibility, increase uncertainty and cause changes in the structures and internal processes of organisations (Al-Mawali and Am 2016; Baines and Langfield-Smith 2003; Chenhall and Langfield-Smith 1998a; Lal and Hassel 1998; Otley 2016; Umanath 2003; Yalcin 2012). The recent COVID-19 pandemic crisis and war in Ukraine have also greatly increased environmental uncertainty (Prohorovs 2022). To cope with these changes and environmental uncertainty, decision-makers are faced with more management accounting information (hereafter only accounting information or information) needs (Afifa and Saleh 2021; Chenhall and Langfield-Smith 1998b; Chenhall and Morris 1986; Pires and Alves 2022) enhancing the importance of information quality (Pedroso and Gomes 2020). That is, they need accurate accounting information to make strategic decisions (Adeniran and Obembe 2020; Oyewo 2021).

As well established in the literature, decision making is one of the principal rationales for management accounting (e.g., Hall 2010; Nielsen et al. 2015; Saukkonen et al. 2018;

Simon 1955). In other words, one of the main functions of management accounting is assisting managers in decision-making processes and providing them with the information that they need (Saukkonen et al. 2018). This information is critical not only to decision facilitating (i.e., information guides decisions and managerial action) but also to decision influencing (i.e., information is used for motivating and controlling managers and employees) (Demski and Feltman 1976; van Veen-Dirks 2010). To be used effectively for both decision facilitating and decision influencing, management accounting information should fit the organisational context and also reflect the roles, responsibilities, and values of all actors that take part in decision-making processes (Saukkonen et al. 2018).

To meet decision-makers' needs, organisations adapt and develop their management accounting systems, which refer to the systematic and interdependent use of practices to support decision making (Chenhall 2003; Grabner and Moers 2013), implementing more contemporary MAPs (e.g., activity-based costing/management, benchmarking, balanced scorecard) that, in addition to traditional accounting information, provide non-financial, external, and future-oriented information and timely information (Abdel-Kader and Luther 2008; Chenhall and Langfield-Smith 1998a, 1998c; Chenhall and Morris 1986; Dahal 2021; Löfsten and Lindelöf 2005). This information is critical to decision-makers acting and making the best possible decisions (Al-Mawali and Am 2016; Oyewo 2021; Visedsun and Terdpaopong 2021) and effectively and efficiently managing the organisations' strategies and operations (Thuan et al. 2022).

In this context, the circumstances of each organisation and the contingency factors determine how best to organise the set of MAPs implemented and adapt it to the accounting information needs (Abdel-Kader and Luther 2008; Cadez and Guilding 2008; Chenhall 2003; Otley 2016; Reid and Smith 2000). In line with the contingency theory, it is important to adjust the implementation and usage of MAPs to contingent factors such as environmental uncertainty, which is not always easy (Otley 2016; Umanath 2003). Whenever this adequacy is achieved, decision-makers are more satisfied with the information obtained (Chenhall 2003; Fry and Smith 1987; Haldma and Lääts 2002; Nicolaou 2000; Tillema 2005). In other words, when the MAPs provide the information that decision-makers need and require, they are satisfied with the information given and with the set of practices implemented (Thuan et al. 2022).

The influence of environmental uncertainty on accounting information characteristics and MAPs has been analysed in different ways. Some studies investigate the relationship between the environmental uncertainty factors and the usefulness or relevance of accounting information (e.g., Al-Mawali and Am 2016; Chenhall and Morris 1986; Pires and Alves 2022). Others directly analyse the relationship between environmental uncertainty and MAPs usage (e.g., Abdel-Kader and Luther 2008; Baines and Langfield-Smith 2003; Löfsten and Lindelöf 2005; Oyewo 2022). However, if the usefulness of accounting information is associated with environmental uncertainty, and if MAPs provide accounting information for decision making, what is the relationship between environmental uncertainty, accounting information usefulness, and MAPs usage? In what circumstances is satisfaction with the accounting information provided by MAPs, which may consider different traditional and/or contemporary MAPs, higher? To the best of our knowledge, no study examines, specifically, the relationships among environmental uncertainty, accounting information usefulness, and MAPs usage and investigates the influence of these relationships on decision-makers' satisfaction with the information provided. This paper aims to fill this gap.

Therefore, the goal of this paper is twofold. Firstly, we examine the relationships among environmental uncertainty, broad-scope and timely accounting information usefulness, and (traditional and contemporary) MAPs usage. Secondly, we intend to explore how the congruent fit between environmental uncertainty, broad-scope and timely accounting information usefulness, and (traditional and contemporary) MAPs usage improves decision-maker's satisfaction with accounting information. The congruence approach assumes that the implementation and usage of MAPs depend on the context without analysing whether this relationship influences managerial or organisational performance (Gerdin and Greve 2004, 2008).

In this study, a quantitative research approach is adopted, and the ontological and epistemological assumptions are compatible with the positivist research paradigm (Hopper and Powell 1985; Modell 2010). The largest manufacturing companies operating in Portugal were surveyed. Data collection was carried out through an online questionnaire survey with an invitation sent via email to the person in charge of management accounting in these companies, as it was considered that this key informant had the appropriate knowledge about the information that was sought to be evidenced. In total, 114 valid responses were obtained, corresponding to a response rate of 23%. Univariate, bivariate, and multivariate techniques were used in the statistical treatment using IBM SPSS Statistics 27.

This study contributes to the literature in several ways. First, providing additional insights concerning the usefulness of broad-scope and timely accounting information for decision-making purposes and the usage of (traditional and contemporary) MAPs under environmental uncertainty, it contributes to the literature on this research topic (e.g., Abdel-Kader and Luther 2008; Afifa and Saleh 2021, 2022; Chenhall and Morris 1986; Chong and Chong 1997; Fisher 1996; Haldma and Lääts 2002; Lal and Hassel 1998; Pires and Alves 2022). In particular, it shows the most useful accounting information to make proper decisions in uncertain contexts (e.g., financial and pandemic crises), which are related to several external changes and unpredictable events. This study also reports evidence on the usage of MAPs that provide this information. Hence, it extends the knowledge of the most suitable accounting information and MAPs for certain situations. Furthermore, our study documents how the congruent fit between environmental uncertainty, accounting information usefulness, and MAPs usage enhances decision-makers' satisfaction with accounting information.

The remainder of this paper is organised as follows. Section 2 summarises the relevant literature and develops our research hypotheses. Section 3 details the research design of this study, including the sample, data collection, and variable measurement. Section 4 presents and discusses the findings. Finally, Section 5 provides the main conclusions, theoretical and practical implications, and limitations of this research, outlining future research opportunities.

## 2. Literature Review and Research Hypotheses

This section comprises three parts. First, the literature regarding the relationship between environmental uncertainty and the usefulness of broad-scope and timely accounting information is reviewed. Second, the literature that relates accounting information usefulness and (traditional and contemporary) MAPs usage is described. Third, a brief review of the literature concerning the congruent fit between environmental uncertainty, accounting information usefulness, and MAP usage, resulting in greater decision-makers' satisfaction with accounting information, is presented.

### 2.1. Environmental Uncertainty and Accounting Information Usefulness

The organisational environment represents the set of external physical and social factors that affect organisations and, therefore, are directly considered in the decision-making process (Duncan 1972; Löfsten and Lindelöf 2005). More specifically, Löfsten and Lindelöf (2005) consider that the organisational environment represents the relevant factors of the organisation's environment that influence the implementation and use of MAPs. These factors can be aggregated into five components that are essential to the functioning of any organisation: customers, suppliers, competitors, socio-political component, and technological component (Duncan 1972). Thus, according to Afifa and Saleh (2022, p. 260), "the environment can be seen as a complex system of interrelated factors with each other, whether economic, market, social, political or cultural factors."

Faced with increasingly global operations, increased competition, permanent and rapid technological changes, and demands for continuous improvement and social and environmental responsibility, the factors of the organisational environment are constantly changing, leading to increased uncertainty (Baines and Langfield-Smith 2003; Chenhall 2003; Latan et al. 2018; Mia

and Clarke 1999; Newkirk and Lederer 2006; Otley 2016; Yalcin 2012). Environmental uncertainty is related to the lack and speed of accounting information as a factor that constrains actions and decisions (Latan et al. 2018). In this context, the importance of the quality of accounting information for decision making increases (Baines and Langfield-Smith 2003; Boulianne 2007; Dahal 2021; Pedroso and Gomes 2020). When the unpredictability of events is greater (Chenhall and Morris 1986), the need to resort to strategic planning (Baines and Langfield-Smith 2003; Newkirk and Lederer 2006) and the introduction of action plans to respond to threats and opportunities increase (Mia and Clarke 1999).

Accounting information gains relevance to dealing with environmental uncertainty (Lal and Hassel 1998; Pires and Alves 2022), as decision-makers need more information in uncertain environments to develop, implement, and monitor organisational strategies (Baines and Langfield-Smith 2003). To address these challenges, traditional accounting information is considered unhelpful and insufficient because it is mainly financial, internal, historical, too aggregated, and not timely (Chenhall and Langfield-Smith 1998c; Johnson and Kaplan 1987). This information does not adequately assess the efficiency of organisations concerned with quality, continuous improvement, and customer satisfaction (Baines and Langfield-Smith 2003).

In situations of greater environmental uncertainty, decision-makers attribute more usefulness to broad-scope information (Afifa and Saleh 2021; Boulianne 2007; Chenhall and Morris 1986; Chong and Chong 1997; Lal and Hassel 1998; Pires and Alves 2022; Yalcin 2012). The scope of accounting information refers to the dimensions of quantification, focus, and time horizon, and broad-scope information comprises, as referred to above, non-financial, external, and future-oriented information (Chenhall and Morris 1986; Le et al. 2020). This allows for making the best decisions in the current conditions of the organisational environment (Adeniran and Obembe 2020; Baines and Langfield-Smith 2003; Oyewo 2021). With the increase in environmental uncertainty, the usefulness of timely accounting information for decision making also increases (Afifa and Saleh 2022; Chenhall and Morris 1986; Fisher 1996).

Therefore, accounting information with these characteristics allows for more appropriate and faster decision making. That is, in situations of high environmental uncertainty, sophisticated accounting information can help decision-makers enhance decision quality, given that this information provides additional alternatives and solutions (Al-Hattami 2022; Latan et al. 2018). Thus, the following research hypotheses are proposed:

**Hypothesis 1a.** *There is a positive relationship between environmental uncertainty and the usefulness of broad-scope accounting information.*

**Hypothesis 1b.** *There is a positive relationship between environmental uncertainty and the usefulness of timely accounting information.*

### 2.2. Accounting Information Usefulness and MAPs

As previously mentioned, permanent changes in the organisational environment increase uncertainty and the usefulness of accounting information for decision making. As a result, there is a need to implement contemporary MAPs to provide broad-scope and timely information, which allows evaluation of the performance of different areas of the organisation and makes the best decisions promptly (Al-Mawali and Am 2016; Baines and Langfield-Smith 2003; Haldma and Lääts 2002; Löfsten and Lindelöf 2005; Oyewo 2021; Visedsun and Terdpaopong 2021). Particularly in the current context, decision-makers use distinct kinds of accounting information and timely information to make effective decisions (Al-Hattami 2022). This information can be used by decision-makers to reduce environmental uncertainty in decision making and also to plan and control activities (Lutfi et al. 2020).

Traditional MAPs (e.g., budgeting systems for planning and controlling costs, standard costing, and variance analysis) primarily provide financial and internal information, with the

main purpose of determining and controlling costs (Abdel-Kader and Luther 2008; Chenhall and Langfield-Smith 1998a; Johnson and Kaplan 1987). These practices do not show the benefits of changes in organisations and the coordination of relationships with the outside world and do not contribute to strategic planning (Chenhall and Langfield-Smith 1998c; Coad 1999; Baines and Langfield-Smith 2003). Contemporary MAPs (e.g., activity-based costing, benchmarking, value chain analysis, and balanced scorecard), in turn, combine financial and non-financial information, internal and external information, and historical and future-oriented information (Baines and Langfield-Smith 2003; Boulianne 2007; Chenhall and Langfield-Smith 1998b, 1998c). These MAPs are useful for strategy development and implementation, and they are critical for achieving long-term organisational goals (Oyewo 2022).

With the increasing need and usefulness of broad-scope and timely information, organisations tend to implement and use contemporary MAPs to provide this information (Abdel-Kader and Luther 2008; Boulianne 2007; Chenhall and Langfield-Smith 1998a; Dick-Forde et al. 2007; Tillema 2005). In this sense, the usefulness of broad-scope and timely accounting information is expected to be positively associated with contemporary MAPs usage.

Following the discussion above, the following research hypotheses are formulated:

**Hypothesis 2a.** *There is a positive relationship between the usefulness of broad-scope accounting information and contemporary MAPs usage.*

**Hypothesis 2b.** *There is a positive relationship between the usefulness of timely accounting information and contemporary MAPs usage.*

Although some authors claim that organisations implement and use contemporary MAPs to satisfy information needs for decision making (e.g., Abdel-Kader and Luther 2008; Boulianne 2007; Dahal 2022), several studies show that traditional MAPs are used more than contemporary ones (e.g., Abdel-Kader and Luther 2006; Chenhall and Langfield-Smith 1998a; Dahal 2021; Dick-Forde et al. 2007; Joshi 2001; Sulaiman et al. 2004; Yalcin 2012). Some of these studies report that traditional MAPs, such as budgets, reach usage levels close to 100%, while contemporary MAPs, for the most part, do not reach usage levels above 50% (e.g., Abdel-Kader and Luther 2006; Boukr et al. 2021; Chenhall and Langfield-Smith 1998a; Dick-Forde et al. 2007; Yalcin 2012). Thus, decision-makers continue to rely on the information provided by traditional MAPs. Besides being more widely used, these practices are also perceived as more important than contemporary MAPs, and their usage is recognised as having more benefits (e.g., Abdel-Kader and Luther 2008; Chenhall and Langfield-Smith 1998a, 1998c; Hyvönen 2005; Joshi 2001). This may occur because the usefulness of financial information for decision-making purposes continues to outstrip that of non-financial information, although both can be seen as complementary (Pires and Alves 2022). As such, the following research hypothesis is proposed:

**Hypothesis 3.** *Traditional MAPs usage is greater than contemporary MAPs usage.*

*2.3. The Influence of the Fit between Environmental Uncertainty, Accounting Information Usefulness, and MAPs on Satisfaction with Accounting Information*

There are several contingent factors (external and internal to the organisation, such as environmental uncertainty, technology, structure, strategy, culture, organisational dimension, and national culture) that influence MAPs (Abdel-Kader and Luther 2008; Baines and Langfield-Smith 2003; Boukr et al. 2021; Chenhall 2003; Chenhall and Morris 1986; Mia and Clarke 1999; Oyewo 2022). Indeed, there is a need to adjust their implementation and usage to provide the information decision-makers need for the planning, control, and decision-making process. Organisations should implement and use contemporary MAPs to keep pace with the growing environmental uncertainty that influences information needs;

organisational structure, strategy, and operations; and, in turn, performance (Afifa and Saleh 2022). The implementation and usage of MAPs should be context-dependent in order to reduce risks and improve competitive advantage and performance (Afifa and Saleh 2022; Dahal 2022).

In this context and following contingency theory, there is no ideal or universally appropriate set of MAPs to be used by all organisations and all circumstances (Afifa and Saleh 2022; Otley 2016; Oyewo 2022; Reid and Smith 2000; Umanath 2003). However, whenever the implementation and usage of MAPs are adapted to contingent factors such as environmental uncertainty, satisfaction with the accounting information provided increases (e.g., Chenhall 2003; Fry and Smith 1987; Haldma and Lääts 2002; Nicolaou 2000; Tillema 2005). When decision-makers' satisfaction with accounting information increases, decision-making effectiveness also increases, which can assist organisations in boosting their chances of survival and success (Al-Hattami 2022). MAPs that provide required information might lead to greater decision makers' satisfaction with accounting information which, in turn, positively influences the overall operations and performance of an organisation (Lutfi et al. 2020). Therefore, the following research hypothesis is proposed:

**Hypothesis 4.** *The fit between environmental uncertainty, accounting information usefulness, and MAPs usage improves decision-makers' satisfaction with accounting information.*

## 3. Research Design

This section encompasses three parts. First, the research approach and data collection methods are described. Second, the sample and data collection procedure are presented. Third, the variables and their measurement are clarified.

### 3.1. Approach and Data Collection

A quantitative approach was adopted to achieve the defined research goals, and the positivist research paradigm was pursued (Cooper and Schindler 2014; Hopper and Powell 1985; Modell 2010; Smith 2003). The type of evidence intended, which is needed to fulfil the research goals, is compatible with an objective conception of reality, seen as external to the researcher. Furthermore, the defined purposes are neither consistent with a subjective view of reality, associated with the interpretative paradigm, nor with the analysis of sources of social and organisational conflict, associated with the critical investigation paradigm (Hopper and Powell 1985; Modell 2010).

A questionnaire survey was used to collect data, essentially because it allowed to reach a large number of respondents and to test the formulated hypotheses (Abdel-Kader and Luther 2008; Gillham 2008; Smith 2003; Stockemer 2019). A structured questionnaire was used (Gillham 2008; Stockemer 2019), as in several previous studies (e.g., Abdel-Kader and Luther 2008; Baines and Langfield-Smith 2003; Cadez and Guilding 2008; Pires and Alves 2022). The questions were arranged into some sections to create a common thread between them and to facilitate asking questions aligned with the research questions.

To ensure that the questionnaire clearly accomplished its objectives, a pre-test was performed. We sent experts (i.e., three academics working in management and accounting departments in Portuguese higher education institutions and five managers of five large manufacturing companies operating in Portugal) a draft of the questionnaire for review. The goals of the pre-test were threefold: (i) to assess the understanding of the design and content of the initial questionnaire, (ii) to determine the time needed to complete the questionnaire, and (iii) to analyse the appropriateness of the subject matter for the population under study (Cooper and Schindler 2014; Stockemer 2019). Based on the feedback, minor adjustments were made in terms of structure and wording to produce the final questionnaire.

For its application, as in several previous studies (e.g., Cadez and Guilding 2008; Dick-Forde et al. 2007; Latan et al. 2018; Pedroso and Gomes 2020; Thuan et al. 2022), the invitation was made via email, and the response via an online platform. Online question-

naires, compared with traditional questionnaires (e.g., face-to-face, telephone, and mail-in), have four main advantages: (i) greater versatility and flexibility to use different types of questions and measurement scales, (ii) faster data collection, (iii) lower cost of administration, and (iv) easier access to large populations (Cooper and Schindler 2014; Hoonakker and Carayon 2009; Stockemer 2019). Furthermore, there are no significant differences between traditional and online questionnaires regarding the response rate, item nonresponse, and nature of the data (Hoonakker and Carayon 2009; Loomis and Paterson 2018).

### 3.2. Sample and Procedure

Once the elaboration of the questionnaire was completed, the study's target population was selected. The National Institute of Statistics (NIS) was asked to list the 500 largest companies (according to turnover) operating in Portugal in the manufacturing industry. The selection of the largest manufacturing companies is justified by the fact that several studies reveal that larger companies have more information needs for the decision-making process, and they are more likely to implement MAPs (e.g., Abdel-Kader and Luther 2008; Boukr et al. 2021; Chenhall 2003; Chenhall and Langfield-Smith 1998a; Haldma and Lääts 2002). These practices are critical for manufacturing companies to control their operations and enhance performance (Dahal 2021, 2022). In addition, most of the studies reviewed also investigated large manufacturing companies (e.g., Baines and Langfield-Smith 2003; Cadez and Guilding 2012; Oyewo 2021, 2022; Visedsun and Terdpaopong 2021; Yalcin 2012). For the comparison of results with these studies to be possible and more rigorous, it was considered important that the target population had similar features.

The questionnaire was sent to the person responsible for the management accounting of the companies that constitute the object of the study. Once this process was completed, 119 questionnaires were received, and 5 were excluded because they were filled out incorrectly (Gillham 2008). A total of 114 responses were considered usable, corresponding to a response rate of 23%, which is in line with the response rate of previous studies (e.g., Latan et al. 2018; Pedroso and Gomes 2020; Visedsun and Terdpaopong 2021).

The characteristics of the responding organisations (turnover and number of employees) and the respondents are summarised in Table 1. It can be seen that 49% of the organisations have a turnover of more than EUR 55 million, and 57% have more than 250 workers. Regarding respondents, 50% of them assume the administrative/financial management position, and the rest have the function of management controller, accountant, administrator/manager, or another function such as economist or assistant financial management. Approximately two-thirds (62.28%) of the respondents have held their current position for more than four years.

**Table 1.** Characteristics of sample and respondents.

| Description | Quantity (%) |
|---|---|
| Size (turnover EUR): | |
| Less than 25 M | 13 (11.40%) |
| 25 M–35 M | 21 (18.42%) |
| 35 M–55 M | 24 (21.05%) |
| 55 M–90 M | 22 (29.82%) |
| More than 90 M | 34 (29.82%) |
| Size (number of employees): | |
| Less than 100 | 15 (13.16%) |
| 100–249 | 34 (29.82%) |
| 250–500 | 38 (33.33%) |
| More than 500 | 27 (23.68%) |

**Table 1.** *Cont.*

| Description | Quantity (%) |
|---|---|
| Current position of respondents: | |
| Managing/financial director | 57 (50.00%) |
| Management controller | 22 (19.30%) |
| Accountant | 21 (18.42%) |
| Administrator/manager | 7 (6.14%) |
| Other | 7 (6.14%) |
| Tenure function (years): | |
| Less than 1 | 10 (8.77%) |
| 1–4 | 33 (28.95%) |
| 5–9 | 30 (26.32%) |
| 10–15 | 20 (17.54%) |
| 16–20 | 16 (14.04%) |
| More than 20 | 5 (4.39%) |

*3.3. Variables Measurement*

Measures developed in previous studies were used or adapted to measure the variables needed to test the formulated research hypotheses in Section 2. To ensure greater confidence and validity of the information collected (Chenhall and Morris 1986; Oyewo 2021), using measures tested in other studies contributes to a more rigorous comparison of results (Chenhall 2003). Below we describe how the variables were operationalised. Survey items are reported in Appendix A.

Environmental uncertainty (EU). We measured the perceived environmental uncertainty using 12 items adapted from the measurement instrument of Newkirk and Lederer (2006), which is a modified version of Teo and King (1997). In addition, we added two items from Löfsten and Lindelöf (2005) (i.e., intensity of research and development; and legal, political, and economic constraints) and one item suggested by Chenhall (2003) (i.e., requirements regarding environmental and social responsibility). Respondents were asked to indicate on a five-point scale (1 = strongly disagree, 5 = strongly agree) their perception about 15 items related to sources of uncertainty. Thus, higher scores indicate more environmental uncertainty. The scores of the items were averaged to determine an overall score for the perceived environmental uncertainty (as reported below, 2 of the 15 items were not included in the overall score). A similar approach was used in previous studies (e.g., Abdel-Kader and Luther 2008; Oyewo 2022).

Broad-scope accounting information (BSAI). To measure the usefulness of broad-scope accounting information, we followed previous studies (e.g., Afifa and Saleh 2021, 2022; Boulianne 2007; Lal and Hassel 1998; Le et al. 2020; Mia and Winata 2008) and used the measurement instrument developed by Chenhall and Morris (1986), which comprises six items. Respondents were requested to indicate on a five-point scale (1 = not useful; 5 = very useful) the usefulness of six items of non-financial, external, and future-oriented accounting information for decision making. In this way, higher scores show the great usefulness of these items of accounting information for decision-making purposes. Rating on these items was averaged to determine an overall score for the usefulness of broad-scope information, similarly to prior studies (e.g., Chenhall and Morris 1986; Mia and Winata 2008).

Timely accounting information (TAI). We measured the usefulness of timely accounting information through the measurement instrument developed by Chenhall and Morris (1986), which contains four items, following previous studies (e.g., Afifa and Saleh 2021, 2022; Bouwens and Abernethy 2000; Lal and Hassel 1998; Le et al. 2020). Respondents were asked to indicate on a five-point scale (1 = not useful; 5 = very useful) the usefulness for decision making of four items related to the frequency and speed of reporting accounting information. Therefore, higher scores indicate great usefulness of timely accounting information for decision making. Rating on these items was averaged to determine an overall score for the usefulness

of timely information. A similar approach was used by Bouwens and Abernethy (2000) and Chenhall and Morris (1986).

Traditional and contemporary MAPs usage. To measure the MAPs usage, we followed previous studies (e.g., Abdel-Kader and Luther 2008; Cadez and Guilding 2008; Oyewo 2022), and we asked respondents to indicate on a five-point scale (1 = not at all; 5 = to a great extent) the extent of usage of each MAP from a list of 35 MAPs. This list, which includes 15 traditional and 20 contemporary MAPs, was developed based on previous studies (e.g., Abdel-Kader and Luther 2008; Chenhall and Langfield-Smith 1998a, 1998c; Sulaiman et al. 2004). The respective scores were averaged to construct overall scores for traditional MAPs usage and contemporary MAPs usage, similarly to Abdel-Kader and Luther (2008) and Oyewo (2022). In this case, higher scores show a great usage of MAPs.

Satisfaction with accounting information (SAI). We measured the satisfaction with accounting information using 12 items adapted from Nicolaou (2000). Respondents were required to indicate on a five-point scale (1 = strongly disagree; 5 = strongly agree) their perception of the quality of information outputs provided by the set of MAPs regarding the information content, accuracy, format, ease of use, and timeliness. Thus, higher scores indicate more decision-makers' satisfaction with accounting information. Rating on these items was averaged to determine an overall score for the decision-makers' satisfaction with accounting information.

## 4. Results and Discussion

This section contains two parts. First, a descriptive analysis of the variables is presented. Second, the hypotheses developed in Section 2 are tested.

### 4.1. Descriptive Analysis

The reliability and descriptive statistics of the variables are presented in Table 2. As shown, except for environmental uncertainty, the variables were made up of all the items considered in the questionnaire (as presented in Section 3.3 and listed in Appendix A). The exclusion of two items (i.e., the survival of this organisational unit is currently threatened by scarce supply of labour, and the survival of this organisational unit is currently threatened by scarce supply of materials) from the instrument of environmental uncertainty improves the internal consistency, measured using Cronbach's Alpha. Cronbach's Alpha coefficient analysis shows that all the variables exceed the recommended cut-off point of 0.70, which reflects an acceptable level of reliability (Cooper and Schindler 2014; Nunnally 1967; Smith 2003).

**Table 2.** Reliability and descriptive statistics of variables.

| Variable | Number of Items Used | Cronbach's Alpha | Mean ($n$ = 114) | Standard Deviation |
|---|---|---|---|---|
| EU | 13 | 0.79 | 3.31 | 0.53 |
| BSAI | 6 | 0.87 | 4.03 | 0.69 |
| TAI | 4 | 0.71 | 4.28 | 0.62 |
| TMAPs | 15 | 0.85 | 3.82 | 0.61 |
| CMAPs | 20 | 0.90 | 2.83 | 0.73 |
| SAI | 12 | 0.92 | 4.01 | 0.51 |

The perception of the respondents regarding environmental uncertainty indicates that this level is moderate (3.31) and results, essentially, from the intense competition in terms of prices, quality, and differentiation of products and the high demands in terms of social and environmental responsibility. Thus, factors related to intense competitiveness drive uncertainty (Baines and Langfield-Smith 2003; Mia and Clarke 1999). The same occurs with social and environmental responsibility demands, as defended by Chenhall (2003), which are perceived as quite high. In fact, organisations are currently facing great uncertainty concerning both the ecological and social environment (Latan et al. 2018).

Concerning the usefulness of accounting information for decision making, it appears that decision-makers at large manufacturing companies operating in Portugal attribute a great deal of usefulness to broad-scope and timely accounting information. Future-oriented information, non-financial information about the market and related to production, as well as obtaining information quickly, are considered by decision-makers as very useful. The high usefulness of this information for decision making is justified by the fact that it allows reacting quickly to changes in the environment and making decisions at the right time (Al-Mawali and Am 2016; Boulianne 2007; Chenhall and Morris 1986; Visedsun and Terdpaopong 2021). More traditional accounting information proves to be insufficient, and sometimes even irrelevant, for decision making because it is eminently financial, internal, historical, and too aggregated and becomes available too late (Baines and Langfield-Smith 2003; Chenhall and Langfield-Smith 1998b, 1998c; Johnson and Kaplan 1987).

Table 2 also reveals that the use of traditional MAPs is quite high and higher than the use of contemporary MAPs. The most used practices are the cost centre method, analysis of fixed and variable costs, budgets for controlling costs, analysis of variances, and analysis of results by products, all traditional MAPs (see Appendix B). About contemporary MAPs, similar to the results obtained by Abdel-Kader and Luther (2006), the analysis of results per customer and the non-financial performance measures related to employees and customers are the most used (see Appendix B).

As for satisfaction with the accounting information produced by the MAPs, the results indicated that respondents consider themselves quite satisfied with the information obtained. These results indicate that the implemented MAPs provide information in a useful format, up-to-date and accurate information, and the information necessary for making decisions on time.

### 4.2. Hypotheses Testing

To test the research hypotheses formulated in the second section, bivariate and multi-variate analyses were used. Spearman's correlation coefficients between the variables are presented in Table 3. Eight of the fifteen relationships established appear to be positive and statistically significant ($p$-value < 0.05).

**Table 3.** Spearman correlations.

| Variable | EU | BSAI | TAI | TMAPs | CMAPs | SI |
|---|---|---|---|---|---|---|
| EU | 1 | | | | | |
| BSAI | 0.17 | 1 | | | | |
| TAI | 0.28 ** | 0.37 ** | 1 | | | |
| TMAPs | 0.06 | 0.40 ** | 0.29 ** | 1 | | |
| CMAPs | 0.17 | 0.21 * | 0.24 ** | 0.62 ** | 1 | |
| SAI | −0.12 | 0.11 | −0.00 | 0.33 ** | 0.07 | 1 |

**, * Significant at 1 and 5% (two-tailed), respectively.

Concerning the relationship between environmental uncertainty and accounting information usefulness for decision making, it appears, on the one hand, that there is no statistically significant relationship between uncertainty and the usefulness of broad-scope information. On the other hand, there is a positive and statistically significant association ($p$-value < 0.01) between environmental uncertainty and the usefulness of timely information for decision making. The results are a little surprising, namely the relationship between environmental uncertainty and the usefulness of broad-scope information. Several studies conclude that when great environmental uncertainty is perceived, greater usefulness is attributed to broad-scope information for decision making (e.g., Afifa and Saleh 2021; Boulianne 2007; Chenhall and Morris 1986; Chong and Chong 1997; Lal and Hassel 1998; Pires and Alves 2022; Yalcin 2012). Broad-scope information, which includes non-financial, external, and future-oriented information, makes it possible to deal more adequately with situations that generate uncertainty. Regarding the relationship between environmental

uncertainty and the usefulness of timely information, the results are consistent with the results obtained by Chenhall and Morris (1986), Fisher (1996), and Afifa and Saleh (2022). It is confirmed that timely information is useful in contexts of more uncertainty because it allows for faster action (Chenhall and Morris 1986), contributing to taking advantage of opportunities and combating possible threats. In this context, Hypothesis 1a. is rejected, and Hypothesis 1b. is supported. Although no hypothesis has been established, there is a positive and statistically significant relationship ($p$-value < 0.01) between the usefulness of broad-scope information and the usefulness of timely information for decision making.

As for the relationship between the accounting information usefulness for decision making and the usage of MAPs, there is a positive and statistically significant association ($p$-value < 0.05) between the usefulness of broad-scope and timely information and the usage of contemporary MAPs (Table 3). These results support Hypotheses 2a. and 2b. and allow us to conclude that when greater usefulness is attributed to broad-scope and timely information, there is greater use of contemporary MAPs, in line with the results of previous studies (e.g., Abdel-Kader and Luther 2008; Baines and Langfield-Smith 2003; Boulianne 2007; Haldma and Lääts 2002; Löfsten and Lindelöf 2005; Tillema 2005). Contemporary MAPs, some with a strategic orientation, provide financial, internal, and historical information and, in particular, non-financial, external, and future-oriented information (Baines and Langfield-Smith 2003; Cadez and Guilding 2008, 2012; Guilding et al. 2000).

Interestingly, and although no hypothesis has been established, the relationship between the usefulness of broad-scope and timely information for decision making and the usage of traditional MAPs is also positive and statistically significant ($p$-value < 0.01). In other words, in situations where more usefulness is attributed to the broad-scope and timely information for decision making, there is greater usage of traditional MAPs. This somewhat surprising relationship, given that traditional MAPs provide more traditional information (Abdel-Kader and Luther 2008; Chenhall and Langfield-Smith 1998a; Johnson and Kaplan 1987) may be justified by the fact that there is a strong positive and statistically significant ($p$-value < 0.01) association between the usage of traditional MAPs and the usage of contemporary MAPs. Therefore, when there is greater usage of contemporary MAPs, which provide broad-scope and timely information (Baines and Langfield-Smith 2003; Cadez and Guilding 2008, 2012; Chenhall and Langfield-Smith 1998c; Guilding et al. 2000), there is also greater usage of traditional MAPs. That is, it seems that traditional and contemporary MAPs are used in a complementary way.

However, the usage of traditional MAPs and the usage of contemporary MAPs are not the same. From the analysis of Table 4, which summarises the results of the non-parametric Wilcoxon test, it can be seen that there is a statistically significant difference ($p$-value < 0.01) between the usage of traditional MAPs and the usage of contemporary MAPs. The analysis of the median usage of traditional MAPs (3.93) and the median usage of contemporary MAPs (2.80) shows that traditional MAPs are used more than contemporary MAPs. This confirms the results of several previous studies (e.g., Abdel-Kader and Luther 2006; Chenhall and Langfield-Smith 1998a; Dahal 2021; Dick-Forde et al. 2007; Joshi 2001; Yalcin 2012). Although several advantages are attributed to the usage of contemporary MAPs because they provide non-financial information oriented toward the outside and toward the future, which contributes to performance evaluation at various levels, to strategic planning, and to obtaining competitive advantages (Abdel-Kader and Luther 2008; Baines and Langfield-Smith 2003; Cadez and Guilding 2008; Guilding et al. 2000; Tillema 2005), traditional MAPs continue to be more widely used by companies. In this context, the results obtained support Hypothesis 3.

**Table 4.** Wilcoxon test: usage of traditional and contemporary MAPs.

| Description | Traditional and Contemporary MAPs |
|---|---|
| Z | −9.22 |
| Asymp. Sig. (two-tailed) | 0.00 |

To test the congruent fit (Gerdin and Greve 2004, 2008), a cluster analysis was carried out, following a widely recommended two-step cluster analysis procedure which has been applied in previous studies (e.g., Cadez and Guilding 2012; Chenhall and Langfield-Smith 1998c). Initially, a hierarchical cluster analysis was performed using the Wards method for grouping cases and squared Euclidean distance as a measure of dissimilarity between cases. With the analysis of the dendrogram, the retention of six clusters was considered a viable solution. The classification of each case in the six retained clusters was refined using the non-hierarchical *K*-Means procedure. The centres of the clusters concerning each variable are shown in Table 5, with the *F*-test for the variables considered.

**Table 5.** Mean scores of variables within clusters (ranking of variables across clusters).

| | Clusters | | | | | | | |
|---|---|---|---|---|---|---|---|---|
| | **C1** | **C2** | **C3** | **C4** | **C5** | **C6** | | |
| *N* | 9 | 17 | 14 | 31 | 24 | 19 | *F*-Test | *p* |
| EU | 3.88 (1) | 2.69 (6) | 3.24 (4) | 3.46 (3) | 3.22 (5) | 3.54 (2) | 12.08 | 0.01 |
| BSAI | 4.78 (1) | 4.37 (2) | 3.13 (6) | 4.24 (4) | 3.51 (5) | 4.37 (2) | 23.11 | 0.01 |
| TAI | 4.61 (3) | 3.97 (5) | 3.32 (6) | 4.69 (2) | 4.06 (4) | 4.70 (1) | 29.07 | 0.01 |
| TMAPs | 4.55 (1) | 4.12 (3) | 3.10 (6) | 4.26 (2) | 3.67 (4) | 3.23 (5) | 38.65 | 0.01 |
| CMAPs | 4.12 (1) | 2.46 (4) | 2.02 (6) | 3.34 (2) | 2.96 (3) | 2.19 (5) | 54.01 | 0.01 |
| SAI | 4.44 (1) | 4.38 (2) | 4.15 (3) | 3.93 (4) | 3.87 (5) | 3.64 (6) | 7.28 | 0.01 |

The clusters were organised in descending order of the degree of satisfaction with the information the MAPs produce. Although the variable related to satisfaction with information has contributed little to differentiate the clusters, there are some differences, namely between the first three and the last cluster. These differences make it possible to perceive that it is important to adapt the use of MAPs to the usefulness of broad-scope and timely information for decision making, which, in turn, is partly related to environmental uncertainty.

Cluster 1 (C1) has the highest level of satisfaction with the accounting information from all clusters. It records the greater usage of traditional and contemporary MAPs, the greater usefulness of broad-scope information for decision making, and the higher level of environmental uncertainty. The usefulness of timely information, although not the highest of all clusters, is very high and indicative of the importance of timely information for decision making. It is verified, therefore, that relations between the variables are consistent. For some environmental uncertainty, more usefulness is given to the broad-scope and timely information for decision making. Consequently, there is greater usage of traditional and contemporary MAPs, which are complementary, as shown in Table 3.

Satisfaction with information in cluster 2 (C2) is very close to that of cluster 1 (C1). However, the setup is quite different. This cluster registers the lowest environmental uncertainty, but it shows a lot of usefulness of broad-scope and timely information for decision making and high use of traditional MAPs. Thus, there is a lack of consistency between the contingent factor related to environmental uncertainty, which is reduced, and the broad-scope and timely information, which is considered very useful for decision making. However, there is some consistency between the usefulness of the information and the use of MAPs, which may explain the high level of satisfaction with the accounting information produced.

In cluster 3 (C3), respondents also consider themselves quite satisfied with the information produced by the MAPs. For a moderate level of environmental uncertainty, little usefulness is given to the broad-scope and timely information, and the use of MAPs is low. Additionally, in this cluster, consistent relationships between the variables are recorded. For lower levels of environmental uncertainty, less usefulness is attributed to accounting information, and MAPs are less used. This means that even when traditional and contemporary MAPs are little used, there can be a high level of satisfaction with the

accounting information provided since the use of MAPs is consistent with the usefulness of accounting information for decision making.

With an intermediate level of satisfaction with the accounting information provided, cluster 4 (C4) registers some environmental uncertainty. The usefulness of broad-scope and timely information for decision making is high, and the usage of MAPs, namely traditional MAPs, is also high. Thus, relations with some consistency between the variables are verified. For some environmental uncertainty, a high level of usefulness is attributed to accounting information for decision making. The usefulness of timely information is higher than that of cluster 1 (C1) and similar to the highest of all clusters. Associated with this high usefulness of accounting information, the usage of MAPs is among the highest of all clusters. Considering the configuration of cluster 1 (C1) and the existing positive association between the usefulness of timely information and the usage of MAPs, it is considered that satisfaction with accounting information is not higher because the usefulness attributed to timely information requires a higher use of traditional and contemporary MAPs and closer to the levels recorded in cluster 1 (C1).

Satisfaction with accounting information in cluster 5 (C5) is very close to that of cluster 4 (C4). The consistency of the relationships between the variables is also similar, although the variables have lower central values. For a very moderate environmental uncertainty, some usefulness is attributed to information for decision making (with timely information being quite useful), and the usage of MAPs is intermediate in the case of traditional ones and reduced in the case of contemporary ones. In this context, and considering the usefulness attributed to timely information, greater usage of MAPs increases the level of satisfaction with accounting information.

Cluster 6 (C6) registers the lowest level of satisfaction with accounting information of all clusters. For one of the highest levels of environmental uncertainty (albeit moderate), it registers, similarly to cluster 1 (C1), much usefulness of broad-scope and timely information for decision making. However, the usage of traditional and contemporary MAPs is more limited. There seems to be some consistency between some variables (the environmental uncertainty and the usefulness of the accounting information) but a lack of consistency between others (the usefulness of the accounting information and the usage of MAPs). In this context, when a lot of usefulness is attributed to accounting information for decision making, high usage of MAPs is fundamental. Otherwise, the satisfaction level with the accounting information is lower.

In short, a consistent relationship between environmental uncertainty, the usefulness of broad-scope and timely information for decision making, and the usage of traditional and contemporary MAPs will positively influence satisfaction with the information the MAPs provide. In the absence of consistency, the level of satisfaction will be lower. Clusters 1 (C1), 3 (C3), and 6 (C6) show this more clearly. However, there is some consistency between the usefulness of the information and the usage of MAPs in clusters 2 (C2) and 4 (C4). For situations with higher environmental uncertainty and moderate usefulness of broad-scope and timely information, a high level of usage of traditional and contemporary MAPs (which complement each other) is fundamental for a high level of satisfaction with accounting information (see C1). If the usage of MAPs is reduced, satisfaction with accounting information is also lower (see C6). In situations of more moderate environmental uncertainty and less usefulness of broad-scope and timely information, reduced usage of MAPs also translates into good satisfaction with accounting information (see C3). Thus, the results show that it is essential to adapt the MAPs usage and usefulness of broad-scope and timely information to contingent factors. These results are consistent with the arguments and conclusions of other authors (e.g., Chenhall 2003; Fry and Smith 1987; Nicolaou 2000; Otley 2016; Tillema 2005) and support Hypothesis 4.

## 5. Conclusions

This study aimed to examine the relationships among environmental uncertainty, the usefulness of broad-scope and timely information, and the usage of traditional and

contemporary MAPs. Moreover, it sought to explore how these relationships influence decision-makers' satisfaction with accounting information. From the literature review, four hypotheses were formulated. Survey data to test them were obtained through an online questionnaire from 114 large manufacturing companies operating in Portugal.

It was found that the environment of the large manufacturing companies operating in Portugal is somewhat uncertain, resulting from factors such as intense competitiveness in terms of prices and high demands in terms of social and environmental responsibility. The decision-makers of these companies attribute a lot of usefulness to broad-scope information for decision making, namely future-oriented information and non-financial information related to the market and production. They also attribute great usefulness to timely information for decision making. Contrary to previous literature and established in Hypothesis 1a, no positive relationship was found between environmental uncertainty and the usefulness of broad-scope information for decision making. On the other hand, a positive relationship between environmental uncertainty and the usefulness of timely information proposed in Hypothesis 1b was confirmed. This result shows the need for decision-makers to obtain accounting information frequently and without delays (Pedroso and Gomes 2020) in order to respond quickly to challenges, mitigate risks, and face changes in a growing competitive environment and market (Le et al. 2020).

As expected in Hypothesis 2a, the results suggest that the greater the usefulness of broad-scope information, the greater the usage of contemporary MAPs. The same is true regarding the usefulness of timely information and contemporary MAPs usage, in line with our Hypothesis 2b. However, although there is greater usage of contemporary MAPs when greater usefulness is attributed to the broad-scope and timely information, it appears that, in global terms, traditional MAPs are the most used, as proposed in Hypothesis 3. Thus, it can be concluded that the large manufacturing companies operating in Portugal use traditional and contemporary MAPs in a complementary way since there is a positive and strong relationship between the usage of traditional MAPs and contemporary MAPs.

Regarding the congruent fit established in Hypothesis 4, it was found that companies that register greater environmental uncertainty attribute great usefulness to broad-scope and timely accounting information for decision making. If companies use, to a large extent, traditional and contemporary MAPs, thus decision-makers present good satisfaction with accounting information. Additionally, companies that register less environmental uncertainty and attribute less usefulness to broad-scope and timely information, even with lower usage of traditional and contemporary MAPs, register good satisfaction with accounting information. The information produced by these MAPs is sufficient for decision making in these contexts. However, companies that register some environmental uncertainty and attribute a lot of usefulness to broad-scope and timely accounting information for decision making, if they make little usage of traditional and contemporary MAPs, register less satisfaction with the accounting information produced. Thus, the importance of adapting traditional and contemporary MAPs usage to the usefulness of broad-scope and timely information for decision making is highlighted, considering the environmental uncertainty. The implementation and use of proper MAPs, which provide adequate information, can improve the effectiveness and efficiency of the decision-making process by creating additional alternatives and solutions, assessing these alternatives, and supporting the choice of the best alternative of action (Al-Hattami 2022; Latan et al. 2018).

The results of this study have significant theoretical and practical implications. From the theoretical point of view, this study contributes to theory by validating, but also questioning/challenging, the findings of previous management accounting contingency studies. On the one hand, the results confirm that environmental uncertainty is positively associated with the usefulness of timely accounting information, and the usage of contemporary MAPs is positively associated with the usefulness of broad-scope and timely information. On the other hand, this study does not find any evidence that supports the positive relationship between environmental uncertainty and broad-scope information. Thus, further research should collect additional evidence on this relationship. In addition, this research also adds

to the existing body of literature, being the first, to the best of our knowledge, to consider simultaneous different features of accounting information, traditional and contemporary MAPs, and satisfaction with accounting information.

From a practical point of view, this study offers valuable insights for practitioners (e.g., management accountants and decision-makers). The results show the need for management accountants and decision-makers to provide and use timely accounting information for decision-making purposes under environmental uncertainty. In addition, the results support the implementation and use, not only of contemporary MAPs but also of traditional MAPs, given their complementary nature, when decision-makers attribute great usefulness to broad-scope and timely accounting information. Management accountants should adjust the implementation of MAPs to the accounting information usefulness for decision making in order to achieve great decision-makers' satisfaction with accounting information.

This study has some limitations that must be considered when interpreting the results obtained. First, the use of the questionnaire survey, which restricts the number of questions asked, does not allow for posing new questions to clarify certain detected situations and is not always completed by the most appropriate person. Second, as this is a cross-sectional study, it cannot establish possible causal relationships. One way to fill these gaps could be using qualitative methods, such as conducting interviews and case studies, which, in turn, do not allow the generalisation of results for the studied sample. Third, the knowledge provided by this study is limited to the Portuguese organisational environment and culture and particularly the context of large manufacturing companies. Thus, further research should be performed in other contexts. In addition, future studies should examine which MAPs provide the most useful accounting information to make different types of decisions, that is, operational and strategic decisions, in order to guarantee decision-makers' satisfaction with accounting information. Moreover, further research should investigate whether great decision-makers' satisfaction with accounting information is related to better decisions and, in turn, to great performance.

**Author Contributions:** Conceptualization, R.P. and M.-C.G.A.; methodology, R.P.; software, R.P.; validation, R.P., M.-C.G.A. and C.F.; formal analysis, R.P.; investigation, R.P.; resources, R.P.; data curation, R.P., M.-C.G.A. and C.F.; writing—original draft preparation, R.P.; writing—review and editing, R.P., M.-C.G.A. and C.F.; visualization, R.P.; supervision, M.-C.G.A.; project administration, R.P., M.-C.G.A. and C.F.; funding acquisition, M.-C.G.A. and C.F. All authors have read and agreed to the published version of the manuscript.

**Funding:** The research of the author Maria-Ceu Gaspar Alves was supported by National Funds through the FCT (Portuguese Foundation for Science and Technology), I.P., within the scope of the project Ref. UIDB/04630/2020. The research of the author Catarina Fernandes was supported by National Funds through the FCT, I.P., within the scope of the project Ref. UIDB/04105/2020.

**Data Availability Statement:** Not applicable.

**Conflicts of Interest:** The authors declare no conflict of interest.

## Appendix A

### Appendix A.1. Environmental Uncertainty (EU) Instrument

Please indicate to what extent you (dis)agree with the following statements regarding the main sources of environmental uncertainty: (1 = strongly disagree; 5 = strongly agree)

1. Products in our industry become obsolete very quickly.
2. The product technologies in our industry change very quickly.
3. We cannot predict what our competitors are going to do next.
4. We cannot predict when our product demand will change.
5. The research and development in our industry are very intense.
6. In our industry, there is considerable diversity in customers' buying habits.
7. In our industry, there is considerable diversity in nature of competition.
8. In our industry, there is considerable diversity in product lines.

9. The survival of this organisational unit is currently threatened by scarce supply of labour.
10. The survival of this organisational unit is currently threatened by scarce supply of materials.
11. The survival of this organisational unit is currently threatened by tough price competition.
12. The survival of this organisational unit is currently threatened by tough competition in product quality.
13. The survival of this organisational unit is currently threatened by tough competition in product differentiation.
14. The survival of this organisational unit is currently threatened by legal, political, and economic constraints.
15. The survival of this organisational unit is currently threatened by the requirements of environmental and social responsibility.

*Appendix A.2. Broad-Scope Accounting Information (BSAI) Instrument*

Please indicate to what extent are useful the following items of management accounting information for your work-related decision making: (1 = not useful, 5 = very useful)

1. Information that relates to possible future events (if historical information is most suitable for your needs, mark the lower end of the scale).
2. Quantification of the likelihood of future events occurring (e.g., probability estimates).
3. Noneconomic information, such as customer preferences, employee attitudes, labour relations, attitudes of government and consumer bodies, competitive threats, etc.
4. Information on broad factors external to your organisation, such as economic conditions, population growth, technological developments, etc.
5. Nonfinancial information that relates to production such as output rates, scrap levels, machine efficiency, and employee absenteeism (if you find that a financial interpretation of production information is most useful for your needs, please mark the lower end of the scale).
6. Nonfinancial information that relates to market information such as market size, growth share, etc. (if you find that a financial interpretation of marketing information is most useful for your needs, please mark the lower end of the scale).

*Appendix A.3. Timely Accounting Information (TAI) Instrument*

Please indicate to what extent are useful the following items related to the frequency and speed of reporting management accounting information for your work-related decision making: (1 = not useful, 5 = very useful)

1. Requested information to arrive immediately upon request.
2. Information is supplied to you automatically upon its receipt into information systems or as soon as processing is completed.
3. Reports are provided frequently on a systematic, regular basis, e.g., daily reports and weekly reports (for less frequent reporting, mark lower end of scale).
4. There is no delay between an event occurring and relevant information being reported to you.

*Appendix A.4. Traditional and Contemporary MAPs Usage Instrument [(T) Represents Traditional MAPs and (C) Represents Contemporary MAPs]*

Please indicate to what extent the following management accounting practices are used in your organisational unit: (1 = not at all; 5 = to a great extent).

1. Budgeting for controlling costs. (T)
2. Flexible budgeting. (T)
3. Budgeting for short-term planning. (T)
4. Budgeting for long-term planning (T)
5. Zero-based budgeting. (C)

6. Using overhead rates. (T)
7. Cost centres method. (T)
8. Standard costing. (T)
9. Variance analysis. (T)
10. Analysis of variable and fixed costs. (T)
11. Cost–volume–profit analysis. (T)
12. Activity-based costing. (C)
13. Activity-based budgeting. (C)
14. Activity-based management. (C)
15. Performance evaluation based on financial measures. (T)
16. Performance evaluation based on non-financial measures related to operations. (T)
17. Performance evaluation based on non-financial measures related to employees. (C)
18. Performance evaluation based on non-financial measures related to customers. (C)
19. Performance evaluation based on economic value added. (C)
20. Evaluation of social and environmental performance. (C)
21. Evaluation of major capital investments based on payback period and/or accounting rate of return. (T)
22. Evaluation of major capital investments based on discounted cash flow method(s). (T)
23. Evaluating the risk of major capital investment projects using probability analysis or computer simulations. (C)
24. Evaluation of major capital investments based on non-financial aspects. (C)
25. Target costing. (C)
26. Kaizen costing. (C)
27. Costs of quality. (C)
28. Product life cycle analysis. (C)
29. Product profitability analysis. (T)
30. Customer profitability analysis. (C)
31. Value chain analysis. (C)
32. Competitors cost analysis. (C)
33. Analysis of competitive position. (C)
34. Benchmarking. (C)
35. Balanced scorecard. (C)

*Appendix A.5. Satisfaction with Accounting Information (SAI) Instrument*

Please indicate to what extent you (dis)agree with the following statements regarding the set of MAPs in use in your organisational unit: (1 = strongly disagree; 5 = strongly agree)

1. The management accounting information is presented in a useful format.
2. You are satisfied with the accuracy of the set of MAPs in use in your organisational unit.
3. The information provided is clear.
4. The information provided is accurate.
5. The MAPs in use in your organisational unit provide sufficient information.
6. The MAPs in use in your organisational unit provide up-to-date information.
7. You get the management accounting information that you need in time.
8. The MAPs in use in your organisational unit provide the precise information that you need.
9. The information content provided by the MAPs in use in your organisational unit meets your needs.
10. The MAPs in use in your organisational unit provide reports that seem to be just about exactly what you need.
11. The control reports are provided frequently on a systematic, regular basis, e.g., daily and weekly reports.
12. The MAPs in use in your organisational unit provide information useful for the ongoing monitoring of your decisions and actions.

## Appendix B.

**Table A1.** Classification and descriptive statistics of MAPs.

| MAPs | Descriptive Statistics | | | Rank (Mean) |
|---|---|---|---|---|
| | **Mean** | **Mode** | **Standard Deviation** | |
| **Traditional MAPs** | | | | |
| Cost centres method | 4.51 | 5 | 0.72 | 1 |
| Analysis of variable and fixed costs | 4.35 | 5 | 0.88 | 2 |
| Budgeting for controlling costs | 4.31 | 5 | 0.96 | 3 |
| Variance analysis | 4.15 | 5 | 1.04 | 4 |
| Product profitability analysis | 3.94 | 5 | 1.07 | 5 |
| Using overhead rates | 3.92 | 5 | 1.03 | 6 |
| Performance evaluation based on financial measures | 3.89 | 4 | 0.93 | 7 |
| Budgeting for long-term planning | 3.82 | 4 | 1.12 | 8 |
| Budgeting for short-term planning | 3.77 | 5 | 1.08 | 9 |
| Performance evaluation based on non-financial measures related to operations | 3.67 | 4 | 0.98 | 11 |
| Cost–volume–profit analysis | 3.54 | 3 | 1.08 | 12 |
| Evaluation of major capital investments based on payback period and/or accounting rate of return | 3.51 | 4 | 1.05 | 13 |
| Standard costing | 3.51 | 5 | 1.33 | 13 |
| Evaluation of major capital investments based on discounted cash flows method(s) | 3.39 | 3 | 1.20 | 15 |
| Flexible budgeting | 3.06 | 4 | 1.29 | 22 |
| **Contemporary MAPs** | | | | |
| Customer profitability analysis | 3.68 | 4 | 1.13 | 10 |
| Performance evaluation based on non-financial measures related to employees | 3.38 | 3 | 0.99 | 16 |
| Performance evaluation based on non-financial measures related to customers | 3.38 | 3 | 1.03 | 16 |
| Costs of quality | 3.26 | 3 | 1.25 | 18 |
| Analysis of competitive position | 3.18 | 3 | 1.11 | 19 |
| Benchmarking | 3.16 | 3 | 1.12 | 20 |
| Activity-based costing | 3.10 | 5 | 1.54 | 21 |
| Activity-based management | 3.00 | 1 | 1.51 | 23 |
| Activity-based budgeting | 2.95 | 1 | 1.53 | 24 |
| Performance evaluation based on economic value added | 2.84 | 3 | 1.34 | 25 |
| Evaluation of major capital investments based on non-financial aspects | 2.76 | 3 | 1.08 | 26 |
| Value chain analysis | 2.72 | 3 | 1.17 | 27 |
| Evaluation of social and environmental performance | 2.68 | 3 | 1.09 | 28 |
| Balanced scorecard | 2.68 | 1 | 1.44 | 28 |
| Competitors cost analysis | 2.61 | 3 | 1.04 | 30 |
| Evaluating the risk of major capital investment projects using probability analysis or computer simulations | 2.42 | 1 | 1.24 | 31 |
| Target costing | 2.37 | 1 | 1.29 | 32 |
| Product life cycle analysis | 2.34 | 1 | 1.22 | 33 |
| Zero-based budgeting | 2.29 | 1 | 1.11 | 34 |
| Kaizen costing | 1.94 | 1 | 1.17 | 35 |

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
