# Peer review of "The Usefulness of Accounting Information and Management Accounting Practices under Environmental Uncertainty"

_jrfm, doi:10.3390/jrfm16020102_

Round 1

Reviewer 1 Report

The presented article for review entitled "The Usefulness of Accounting Information and Management 3 Accounting Practices Under Environmental Uncertainty" answers a very interesting research problem. First, the relationship among environmental uncertainty, broad scope, and timely accounting information usefulness, and (traditional and contemporary) management accounting practices (MAPs) usage is examined. Secondly, the Authors intend to explore how these relationships influence decision-makers satisfaction with accounting in8 formation. The research was developed using an online survey among 114 companies. The findings indicate a positive relationship between environmental uncertainty and timely information usefulness and between (broad scope and timely) information usefulness and (traditional and contemporary) MAPs usage. I believe that the results achieved in this study are useful for both theory and practice and have several implications for professionals engaged in MAS implementation and decision-making activities.

Notes to the article:

1. The article contains a very good and up-to-date review of the literature on the subject.

2. The research hypotheses were put forward correctly, although in the reviewer's opinion, one hypothesis can be left and the other two considered as partial hypotheses (this applies to the first and the second hypothesis). The third and fourth hypotheses were correctly constructed.

3. The selection of study procedures was prepared correctly, and the choice of variables was as well - no comments.

4. Statistical analysis results - no comments.

5. I would organize the conclusions by assigning them more clearly to the hypotheses.

I wish you fruitful scientific work!

Author Response

We would like to thank the reviewers for their constructive and helpful comments and suggestions concerning our manuscript. We trust that the way we addressed all the issues will be seen as satisfactory for the reviewers while keeping the focus of the revised manuscript intact and the length reasonable. We hope the paper is now suitable for publication. Notwithstanding, we remain totally at the reviewers’ disposal to further clarify any issues or introduce additional changes.

Changes are marked in red in the revised manuscript. Also, we used red below to distinguish our responses from the reviewers’ valuable comments and suggestions.

Reviewer 1’s Comments/Suggestions and Authors’ Responses

Comment/suggestion #1:The article contains a very good and up-to-date review of the literature on the subject.

We are grateful for the reviewer’s comment.

Comment/suggestion #2:The research hypotheses were put forward correctly, although in the reviewer's opinion, one hypothesis can be left and the other two considered as partial hypotheses (this applies to the first and the second hypothesis). The third and fourth hypotheses were correctly constructed.

We thank the reviewer’s comment. Based on the reviewer’s suggestion, we have reformulated the first and second hypotheses.

Comment/suggestion #3:The selection of study procedures was prepared correctly, and the choice of variables was as well - no comments.

We appreciate the reviewer’s comment.

Comment/suggestion #4:Statistical analysis results - no comments.

We are grateful for the reviewer’s comment.

Comment/suggestion #5:I would organize the conclusions by assigning them more clearly to the hypotheses.

We thank the reviewer’s comment. In accordance with it, we have organised the conclusions in order to assign them more clearly to the research hypotheses.

Once again, thank you very much for your comments and suggestions.

Reviewer 2 Report

The paper under title “The Usefulness of Accounting Information and Management Accounting Practices Under Environmental Uncertainty” examines how the relevance of management accounting information and practices is important for industrial firms under environmental uncertainty. The study uses a questionnaire-quantitative type of analysis on a sample of 114 respondents from Portuguese firms. The paper provides a perception-type of methodology extracting information from specialized employees related to management accounting. The paper could be of interest to accounting researchers and those interested in the usefulness of management accounting information. The paper thought requires some refinements in terms of theoretical justification and contribution to existing literature, thus I propose a revise and resubmit decision, based on the following comments.

1.      Firstly, the terms “MAP” and “MAS” is used interchangeably in the paper and is somehow confusing. Please use the same term throughout the paper. Since the focus of the study is on practices please use a more concrete explanation on the difference between MAS and MAP.

2.      Also, the focus of the paper is on decision making accounting information and specifically management accounting. So, a theoretical background is required on the introduction regarding the relevance of management accounting information for decision making. This needs to be explicitly explained in the introduction. Also, the reference on “accounting information” should be made specific on management accounting information (on the whole paper), since the literature on accounting relevance refers to financial accounting measures also. In order to avoid any confusing to the reader please make clear that you mean management accounting information.

3.      The contribution of the study (page 2) must be connected to existing literature.

4.      On hypothesis 1.2, the timely information is referred to financial or managerial accounting information?

5.      On section 3.3 more details are required on the interpretation of the variables estimated. For example the UE variable higher score means less environmental uncertainty? Please make clear the variable’s estimation process and interpretation.

6.      Have author(s) considered estimating the variables via principal component or factor extraction analysis? At least as a sensitivity test?

7.      On section 3.3 environmental uncertainty is estimated using 14 items, but on table 2 UE is estimated based on 13 items. Please revise if necessary.

8.      Also, on lines 321 what do you mean by “additively combined”? Can author(s) provide a clear explanation on the estimation process?

9.      The appendix should include the items of the questionnaire used for measuring the variables on page 7 and “satisfaction with information” variable on page 8.

10.   On section 4.1, lines 359-360 author(s) mention “As shown, except for environmental uncertainty, the variables were made up of all the items considered in the questionnaire”. Why UE has not included all items? Please explain.

11.   In addition, what type of threshold has been used to judge a “good or very good internal consistency”? A 0.75 perhaps? Please make this clear.

12.   Finally, the conclusion could offer to the readers some directions for future research.

Author Response

We would like to thank the reviewers for their constructive and helpful comments and suggestions concerning our manuscript. We trust that the way we addressed all the issues will be seen as satisfactory for the reviewers while keeping the focus of the revised manuscript intact and the length reasonable. We hope the paper is now suitable for publication. Notwithstanding, we remain totally at the reviewers’ disposal to further clarify any issues or introduce additional changes.

Changes are marked in red in the revised manuscript. Also, we used red below to distinguish our responses from the reviewers’ valuable comments and suggestions.

Reviewer 2’s Comments/Suggestions and Authors’ Responses

Comment/suggestion #1:Firstly, the terms “MAP” and “MAS” is used interchangeably in the paper and is somehow confusing. Please use the same term throughout the paper. Since the focus of the study is on practices please use a more concrete explanation on the difference between MAS and MAP.

We are grateful for the reviewer’s comment.

In the literature, several terms such as management accounting practices (MAPs), management accounting (MA), management accounting systems (MAS), and management control systems (MCS) are not seldom used interchangeably (e.g., Bedford et al., 2016; Chenhall, 2003; Grabner & Moers, 2013; Malmi & Brown, 2008). However, they have different meanings. For instance, according to Chenhall (2003), “MA refers to a collection of practices such as budgeting or product costing” (p. 129). In addition, he notes that “MAS refers to the systematic use of MA to achieve some goal” and “MCS is a broader term that encompasses MAS and also includes other controls such as personal or clan controls” (p. 129). Other authors distinguish between management control as a package and management control as a system (e.g., Bedford et al., 2016; Grabner & Moers, 2013; Malmi & Brown, 2008). Management control as a package means a set of practices, and management control as a system refers to management control practices that are interdependent and, thus, the value of one practice depends on the value of another (Grabner & Moers, 2013). When these systems are designed to assist decision-making, but leave such use unmonitored, they should be termed MAS (Malmi & Brown, 2008).

The package of practices (i.e., MAPs), which Chenhall (2003) label as MA, is the focus of our study. As in previous studies (e.g., Abdel-Kader & Luther, 2008; Sulaiman et al., 2004), these practices are grouped into traditional and contemporary MAPs. When these traditional and contemporary MAPs are interdependent and form a system that supports decision-making then should be labelled as MAS.

Therefore, in accordance with the reviewer’s comment and the discussion above, we have revised the paper to verify the correct use of the terms MAPs and MAS. In addition, we have added a brief explanation of the differences between MAPs and MAS.

Comment/suggestion #2:Also, the focus of the paper is on decision making accounting information and specifically management accounting. So, a theoretical background is required on the introduction regarding the relevance of management accounting information for decision making. This needs to be explicitly explained in the introduction. Also, the reference on “accounting information” should be made specific on management accounting information (on the whole paper), since the literature on accounting relevance refers to financial accounting measures also. In order to avoid any confusing to the reader please make clear that you mean management accounting information.

We thank the reviewer’s comment.

This study focuses on management accounting information provided by MAPs for decision-making purposes. Although we use interchangeably the terms management accounting information, accounting information, and information throughout the paper, at the beginning of the Section 1 «Introduction» of the revised manuscript we state now that accounting information or only information means management accounting information. In accordance with the reviewer’s recommendations, we have also added on Section 1 «Introduction» additional explanations concerning the relevance of management accounting information for decision-making purposes.

Comment/suggestion #3:The contribution of the study (page 2) must be connected to existing literature.

We appreciate the reviewer’s comment. In accordance with it, we have connected the contribution of our study to existing literature. Moreover, we have reinforced the contribution of the study, highlighting, now, the relevance of the congruent fit examined throughout the manuscript on the decision-makers’ satisfaction with accounting information.

Comment/suggestion #4:On hypothesis 1.2, the timely information is referred to financial or managerial accounting information?

We are grateful for the reviewer’s comment.

As stated above, our study focuses on management accounting information provided by MAPs. Although we use interchangeably the terms management accounting information, accounting information, and information throughout the paper, at the beginning of the introduction of the revised manuscript we mention now that accounting information or only information means management accounting information. Thus, hypotheses 1.2. and 2.2., respectively 1b. and 2b in the revised manuscript, refer on timely management accounting information.

Comment/suggestion #5:On section 3.3 more details are required on the interpretation of the variables estimated. For example the UE variable higher score means less environmental uncertainty? Please make clear the variable’s estimation process and interpretation.

We thank the reviewer’s comment.

In this study, we adapted/used measurement instruments for the variables developed as previously used in similar studies. The operationalization of these variables was also performed similarly to previous studies (e.g., Hoque, 2005; Oyewo, 2022). Specifically, higher scores of the items, depending of the variable, indicate more uncertainty, usefulness, usage, or satisfaction. That is, concerning environmental uncertainty variable, higher score means more environmental uncertainty. Regarding the usefulness of broad scope and timely accounting information, higher score means more usefulness of broad scope and timely accounting information for decision-making purposes. In the case of MAPs, higher score means a great usage of these MAPs. Finally, a higher score of the satisfaction witch accounting information means that decision-makers are more satisfied with the information provided by the MAPs implemented and used in their organisations.

Therefore, in accordance with the reviewer’s comment and recommendation, and the discussion above, we have added additional explanations in the Subsection 3.3. «Variables Measurement» in order to improve the interpretation of the variables.

Comment/suggestion #6:Have author(s) considered estimating the variables via principal component or factor extraction analysis? At least as a sensitivity test?

We appreciate the reviewer’s comment.

In fact, in this study we have not estimated the variables via principal component or factor analysis. Since this study is part of a research project, which considers the publication of additional papers, some already in progress, we will incorporate the estimation method suggested by the reviewer in such papers.

In any case, in this study, we followed several steps to guarantee the validity and reliability of the variables. First, as mentioned in the manuscript (Subsection 3.3. «Variables Measurement»), the measurement instruments of this study were adapted from previous studies to achieve internal validity. Second, we carried out a pre-test of the questionnaire to improve the adaptation of these instruments to the context (additional information concerning this pre-test was added in the Subsection 3.1. «Approach and Data Collection»). Third, we examined reliability using Conbrach’s Alpha.

Comment/suggestion #7:On section 3.3 environmental uncertainty is estimated using 14 items, but on table 2 UE is estimated based on 13 items. Please revise if necessary.

We are grateful for the reviewer’s comment.

The initial measurement instrument of perceived environmental uncertainty contained 15 items. However, after analysis of internal consistency, two items were excluded. As a result, the variable of environmental uncertainty was made up of 13 items. Please see the response to Comment/suggestion #10 for additional details.

Comment/suggestion #8:Also, on lines 321 what do you mean by “additively combined”? Can author(s) provide a clear explanation on the estimation process?

We thank the reviewer’s comment.

As mentioned above, the operationalization of the variables included in this study was performed similarly to previous studies (e.g., Henri, 2006; Hoque, 2005; McManus, 2013; Oyewo, 2022). For each variable were summed up (additively combined) the scores of the items included in the estimation of the variables and then the result of this sum was divided by the number of items (averaged). For instance, for the variable timely accounting information usefulness, we summed up the scores of the four items (Item 1 + Item 2 + Item 3 + Item 4) and then we divide them by four. We have revised the explanation on the estimation process and have introduced some changes.

Comment/suggestion #9:The appendix should include the items of the questionnaire used for measuring the variables on page 7 and “satisfaction with information” variable on page 8.

We appreciate the reviewer’s comment. In accordance with the reviewer’s recommendation, we have included in Appendix A the items of the survey instrument used to measure the variables of the study.

Comment/suggestion #10:On section 4.1, lines 359-360 author(s) mention “As shown, except for environmental uncertainty, the variables were made up of all the items considered in the questionnaire”. Why UE has not included all items? Please explain.

We are grateful for the reviewer’s comment.

The initial measurement instrument of the perceived environmental uncertainty contained 15 items, as presented in Subsection 3.3. «Variables Measurement» and listed now in the Appendix A of the revised manuscript: 12 items adapted from the measurement instrument of Newkirk and Lederer (2006), two items adapted from Löfsten and Lindelöf (2005), and one item adapted from Chenhall (2003). However, analysing the internal consistency of these 15 items through Cronbach’s Alpha, we verified that the exclusion of two items (i.e., the survival of this organisational unit is currently threatened by scarce supply of labour and the survival of this organisational unit is currently threatened by scarce supply of materials) improve such internal consistency. As a result, the variable was made up of 12 items.

Thus, in order to clarify the readers, we have added an additional explanation on the Subsection 4.1. «Descriptive Analysis» concerning the exclusion of two items from the initial measurement instrument of environmental uncertainty.

Comment/suggestion #11:In addition, what type of threshold has been used to judge a “good or very good internal consistency”? A 0.75 perhaps? Please make this clear.

We thank the reviewer’s comment.

Previous literature (e.g., Henri & Wouters, 2020; Nguyen & Nguyen, 2021) has used different thresholds, 0.60 or 0.70, to judge reliability as acceptable. Accordingly, in this study we have used the threshold of 0.70. We have now revised the text in order to make this clearer.

Comment/suggestion #12:Finally, the conclusion could offer to the readers some directions for future research.

We appreciate the reviewer’s comment. In accordance with it, at the end of the Section 5. «Conclusions», we have added some opportunities for further research.

Once again, thank you very much for your comments and suggestions.

References

Abdel-Kader, M., & Luther, R. (2008). The impact of firm characteristics on management accounting practices: A UK-based empirical analysis. The British Accounting Review, 40(1), 2–27. https://doi.org/10.1016/J.BAR.2007.11.003

Bedford, D. S., Malmi, T., & Sandelin, M. (2016). Management control effectiveness and strategy: An empirical analysis of packages and systems. Accounting, Organizations and Society, 51, 12–28. https://doi.org/10.1016/J.AOS.2016.04.002

Chenhall, R. H. (2003). Management control systems design within its organizational context: Findings from contingency-based research and directions for the future. Accounting, Organizations and Society, 28(2–3), 127–168. https://doi.org/10.1016/S0361-3682(01)00027-7

Grabner, I., & Moers, F. (2013). Management control as a system or a package? Conceptual and empirical issues. Accounting, Organizations and Society, 38(6–7), 407–419. https://doi.org/10.1016/J.AOS.2013.09.002

Henri, J.-F. (2006). Organizational culture and performance measurement systems. Accounting, Organizations and Society, 31(1), 77–103. https://doi.org/10.1016/j.aos.2004.10.003

Henri, J.-F., & Wouters, M. (2020). Interdependence of management control practices for product innovation: The influence of environmental unpredictability. Accounting, Organizations and Society, 86, 1–14. https://doi.org/10.1016/J.AOS.2019.101073

Hoque, Z. (2005). Linking environmental uncertainty to non-financial performance measures and performance: A research note. British Accounting Review, 37(4), 471–481. https://doi.org/10.1016/j.bar.2005.08.003

Löfsten, H., & Lindelöf, P. (2005). Environmental hostility, strategic orientation and the importance of management accounting—An empirical analysis of new technology-based firms. Technovation, 25(7), 725–738. https://doi.org/10.1016/J.TECHNOVATION.2004.01.007

Malmi, T., & Brown, D. A. (2008). Management control systems as a package—Opportunities, challenges and research directions. Management Accounting Research, 19(4), 287–300. https://doi.org/10.1016/J.MAR.2008.09.003

McManus, L. (2013). Customer accounting and marketing performance measures in the hotel industry: Evidence from Australia. International Journal of Hospitality Management, 33(1), 140–152. https://doi.org/10.1016/j.ijhm.2012.07.007

Newkirk, H. E., & Lederer, A. L. (2006). The effectiveness of strategic information systems planning under environmental uncertainty. Information & Management, 43(4), 481–501. https://doi.org/10.1016/J.IM.2005.12.001

Nguyen, T. M., & Nguyen, T. T. (2021). The application of strategic management accounting: Evidence from the consumer goods industry in Vietnam. Journal of Asian Finance, Economics and Business, 8(10), 139–146. https://doi.org/10.13106/jafeb.2021.vol8.no10.0139

Oyewo, B. (2022). Contextual factors moderating the impact of strategic management accounting on competitive advantage. Journal of Applied Accounting Research, 23(5), 921–949. https://doi.org/10.1108/JAAR-04-2021-0108

Sulaiman, M. B., Ahmad, N. N. N., & Alwi, N. (2004). Management accounting practices in selected Asian countries: A review of the literature. Managerial Auditing Journal, 19(4), 493–508. https://doi.org/10.1108/02686900410530501

Round 2

Reviewer 2 Report

I would like to thank authors for their efforts to address all comments raised. I believe the paper has improved relative to its initial draft. I though propose a final check for any remaining syntax errors and for the consistency and completeness of the reference list. I believe that the paper has reached its publishable form conditional to the two issues mentioned above.